# A Lipidomics- and Transcriptomics-Based Analysis of the Intestine of Genetically Obese (*ob/ob*) and Diabetic (*db/db*) Mice: Links with Inflammation and Gut Microbiota

**DOI:** 10.3390/cells12030411

**Published:** 2023-01-25

**Authors:** Francesco Suriano, Claudia Manca, Nicolas Flamand, Matthias Van Hul, Nathalie M. Delzenne, Cristoforo Silvestri, Patrice D. Cani, Vincenzo Di Marzo

**Affiliations:** 1Metabolism and Nutrition Research Group, Louvain Drug Research Institute (LDRI), UCLouvain, Université Catholique de Louvain, Av. E. Mounier, 73 B1.73.11, 1200 Brussels, Belgium; 2Walloon Excellence in Life Sciences and BIOtechnology (WELBIO), WELBIO Department, WEL Research Institute, Avenue Pasteur, 6, 1300 Wavre, Belgium; 3Quebec Heart and Lung Institute Research Centre, Université Laval, Quebec City, QC G1V 0A6, Canada; 4Institute of Nutrition and Functional Foods, Centre NUTRISS, Université Laval, Quebec City, QC G1V 0A6, Canada; 5Joint International Research Unit for Chemical and Biomolecular Research on the Microbiome and Its Impact on Metabolic Health and Nutrition (JIRU-MicroMeNu) between Université Laval and the Italian Consiglio Nazionale delle Ricerche, Institute of Biomolecular Chemistry, Consiglio Nazionale delle Ricerche, 80078 Pozzuoli, Italy

**Keywords:** lipidomics, transcriptomics, endocannabinoids, enzymes, intestine, obesity, diabetes, inflammation, gut microbiota

## Abstract

Obesity is associated with a cluster of metabolic disorders, chronic low-grade inflammation, altered gut microbiota, increased intestinal permeability, and alterations of the lipid mediators of the expanded endocannabinoid (eCB) signaling system, or endocannabinoidome (eCBome). In the present study, we characterized the profile of the eCBome and related oxylipins in the small and large intestines of genetically obese (*ob/ob*) and diabetic (*db/db*) mice to decipher possible correlations between these mediators and intestinal inflammation and gut microbiota composition. Basal lipid and gene expression profiles, measured by LC/MS-MS-based targeted lipidomics and qPCR transcriptomics, respectively, highlighted a differentially altered intestinal eCBome and oxylipin tone, possibly linked to increased mRNA levels of inflammatory markers in *db/db* mice. In particular, the duodenal levels of several 2-monoacylglycerols and *N*-acylethanolamines were increased and decreased, respectively, in *db/db* mice, which displayed more pronounced intestinal inflammation. To a little extent, these differences were explained by changes in the expression of the corresponding metabolic enzymes. Correlation analyses suggested possible interactions between eCBome/oxylipin mediators, cytokines, and bacterial components and bacterial taxa closely related to intestinal inflammation. Collectively, this study reveals that *db/db* mice present a higher inflammatory state in the intestine as compared to *ob/ob* mice, and that this difference is associated with profound and potentially adaptive or maladaptive, and partly intestinal segment-specific alterations in eCBome and oxylipin signaling. This study opens the way to future investigations on the biological role of several poorly investigated eCBome mediators and oxylipins in the context of obesity and diabetes-induced gut dysbiosis and inflammation.

## 1. Introduction

Overweight and obesity have reached epidemic proportions globally and are one of the leading public health issues of the 21st century [1,2]. Both human and animal studies have underlined the importance of the microorganisms living in our gastrointestinal (GI) tract, collectively referred to as the gut microbiota, in the protection or the onset of many diseases including obesity, type 2 diabetes and intestinal disorders [3,4]. In fact, microbes coevolve with their host in a symbiotic relationship [3,4] and, in physiological conditions, offer many benefits, which include fortifying the intestinal barrier, harvesting energy from undigested and unabsorbed nutrients, defending against pathogens, and regulating host immunity [5,6]. However, gut microbiota perturbations by several environmental (e.g., diet) and intrinsic (e.g., genetic) factors, generally defined as “dysbiosis”, result in impaired gut barrier integrity, which leads to the translocation of bacterial components, such as pro-inflammatory lipopolysaccharides (LPS), into the systemic circulation [7,8]. These phenomena initiate the development of chronic low-grade inflammation (also known as metabolic endotoxemia), thus contributing to insulin resistance in peripheral organs (e.g., liver, adipose tissue) [7,8], and suggesting that an intact intestinal barrier acts as a gatekeeper of our health, crucial to avoid the development of intestinal and extra-intestinal diseases [9].

Among the different systems implicated in regulating the intestinal barrier alongside the gut microbiota and bacterial metabolites is the expanded endocannabinoid (eCB) system—also known as the endocannabinoidome (eCBome)—which includes the eCBs (*N*-arachidonoyl-ethanolamine, AEA, and 2-arachidonoylglycerol, 2-AG), their congeners (the *N*-acylethanolamines, NAEs, and the monoacyl-glycerols, 2-MAGs, respectively), and other eCB-like bioactive lipid mediators, their receptors, and metabolic enzymes [10,11]. The eCBome plays an important role in the maintenance of intestinal homeostasis by regulating permeability and the inflammatory response of the human gut [12,13], as demonstrated by recent studies revealing modifications in the expression of genes encoding the eCBome G protein-coupled receptor 55 (GPR55), cannabinoid receptor 1 and 2 (CB1 and CB2), and the 2-MAG catabolic enzyme monoacylglycerol lipase (MGL), during intestinal inflammation and chronic colitis [14]. Tissue and circulating levels of eCBome molecules can also be altered by dietary factors, as well as by the gut microbiota. Muccioli and colleagues found that mice fed a high-fat diet displayed an altered intestinal microbiota, which leads to an increased intestinal eCB/CB1 tone, resulting in inflammation and incremented plasma LPS levels, augmented intestinal permeability, and dysregulated adipogenesis [12]. On the other hand, it has been shown that the in vivo deletion of the NAE synthesizing enzyme *N*-acylphosphatidylethanolamine phospholipase D, *Napepld,* in white adipocytes or intestinal epithelial cells, induces a shift in gut microbiota composition and related dysmetabolism [15,16]. In antibiotic-treated mice, the levels of the eCB-related bioactive long-chain fatty acid derivatives, i.e., the NAEs and *N*-acylserotonins, were modified in the small intestine [17]. Additionally, the eCBome is directly affected by the presence or absence of gut microbes, as shown in germ-free (GF) mice exhibiting significant modifications in intestinal eCBome mediators, which were partially or completely reversed by fecal microbiota transplantation from donor to age-matched GF mice, thus highlighting the cause-effect relationship between the presence or absence of intestinal microbes and eCBome signaling [18]. 

In the context of obesity, type 2 diabetes (T2D), and inflammation we have previously shown that genetically obese (*ob/ob*) and diabetic (*db/db*) mice, despite exhibiting the same body weight and fat mass, are characterized by a divergent eCBome signaling in two different sites (i.e., liver and adipose tissues depots) that could be related to the etiology or consequences of the different inflammatory tone observed in the two mutant strains [19,20]. As a consequence, the higher circulating LPS levels observed in the *db/db* mice reflect an increased intestinal permeability [19], which might be ascribed to a different inflammatory tone as well as to a distinct eCBome profile in the different intestinal segments. Therefore, in the present study, we aimed at characterizing the eCBome and oxylipin profiles in different intestinal segments of *ob/ob* and *db/db* mice using targeted LC/MS-MS-based lipidomics and qPCR array-based transcriptomics. Additionally, by performing correlative analyses, we sought to determine to what extent specific bacterial components and bacterial taxa could be related to either inflammatory markers or eCBome components. Deciphering the role that the lipidome targeted here plays in intestinal inflammation, per se or in relation to gut microbiota composition, might open up new possibilities for the treatment of metabolic diseases.

## 2. Materials and Methods

### 2.1. Animals

As previously described [19], six-week-old male homozygous *ob/ob* (B6.V-Lepob/ob/JRj) and *db/db* mice (BKS-Lepr/db/db/JOrlRj) and their respective control lean littermates (*n* = 9–10 per experimental group) were used and followed for seven weeks. All mouse experiments were approved by and performed in accordance with the guideline of the local ethics committee (Ethics committee of the Université catholique de Louvain for Animal Experiments specifically approved this study that received the agreement number 2017/UCL/MD/005). Housing conditions were specified by the Belgian Law of 29 May 2013, regarding the protection of laboratory animals (agreement number LA1230314).

### 2.2. Tissues

The duodenum, jejunum, ileum and distal colon used in this study to explore the eCBome tone originated from the same mice extensively phenotyped in Suriano et al., [19]. Briefly, at the end of the experimental period and after 6 h of fasting, mice were anesthetized with isoflurane (Forene, Abbott, Queenborough, Kent, UK). The abdominal cavity was opened and the whole digestive tract was carefully aligned from the stomach up to the colon. Once the mesenteric adipose tissue and the stomach were removed, the full intestinal length was measured. The small intestine was measured from the pyloric sphincter to the ileocecal junction, while the large intestine was measured from the colon-cecum to the colorectal margin. Subsequently, the small intestine (duodenum, jejunum, ileum) and large intestine (colon) were carefully excised and separated. After removal of the lumen content, the gut segments were immediately snap frozen in liquid nitrogen and stored at −80 °C for further analysis. This common procedure was concluded for each mouse within a maximum of 10 min, a time frame that allowed for the preservation of mRNA, and lipid for further analysis. 

### 2.3. Lipid Extraction and HPLC-MS/MS for the Analysis of eCBome Mediators

Lipids were extracted from tissue samples according to the Bligh and Dyer method [21], as previously described [16,20].

The quantification of eCBome-related mediators (Appendix A), was carried out by HPLC interfaced with the electrospray source of a Shimadzu 8050 triple quadrupole mass spectrometer using multiple reaction monitoring in positive ion mode for the compounds and their deuterated homologs. In the case of unsaturated monoacylglycerols, the data are presented as 2-monoacylglycerols (2-MAGs) but represent the combined signals from the 2- and 1(3)-isomers since the latter are most likely generated from the former via acyl migration from the sn-2 to the sn-1 or sn-3 position. 

### 2.4. RNA Isolation, Reverse Transcription and qPCR-Based TaqMan Open Array

Total RNA and cDNA were obtained, and their quantity and quality were assessed as previously described [19]. Sixty-five nanograms of starting RNA were used to evaluate the expression of the 52 eCBome-related genes and four housekeeping genes (Appendix A) as previously described [20]. mRNA expression levels were calculated from duplicate reactions using the 2^−ΔΔCt^ method and are represented as fold change with respect to baseline within each tissue. *Rps13* was used as a reference gene for the eCBome-related receptors and metabolic enzymes, whereas *Rpl19* was chosen as the housekeeping gene for the intestinal inflammatory markers. cDNA was prepared by reverse transcription of 1 µg total RNA using the Goscript RT Mix OligoDT kit (Promega, Leiden, The Netherlands) as previously described [19]. The primer sequences for the inflammation related genes are presented in Appendix A. 

### 2.5. Correlation Analysis

As previously described [20], a correlation analysis between two data sets of variables was performed using the R package ‘psych’ (version 2.1.6). All statistical analyses were performed on RStudio (version 4.1.0, Rstudio Team, Boston, MA, USA).

### 2.6. Statistical Analysis

Data are presented as the mean ± standard error of the mean (S.E.M), as specified in the individual tables and figures. The differences between the groups were determined using a One-Way ANOVA followed by Tukey’s post hoc test on ΔΔCt and on fmol/mg tissue for gene expression levels and mediator levels, respectively. Only statistically significant differences between *ob/ob* and *db/db* mice were reported. The differences between experimental groups were considered statistically significant with a *p* ≤ 0.05 and were represented as follows: * *p* ≤ 0.05, ** *p* ≤ 0.01, *** *p* ≤ 0.005, **** *p* ≤ 0.001. Data were analyzed using GraphPad Prism version 8.00 for Windows (GraphPad Software, San Diego, CA, USA). The presence of outliers was assessed using the Grubbs test.

## 3. Results

### 3.1. Distinct Intestinal Length and Inflammatory Tone in db/db and ob/ob Mice

At necropsy, *db/db* mice had a significant increase in the full intestinal length as compared to *ob/ob* mice (Figure 1A). To investigate whether this change was accompanied by a more pronounced inflammatory tone in the intestine, we investigated the mRNA expression of different pro-inflammatory markers (i.e., *Il6*, *Il1b*, *Tnfa*, *Rorgt2*, and *Il17a*) in three segments of the small intestine (i.e., duodenum, jejunum, and ileum) and in one segment of the large intestine (i.e., distal colon). Although no significant changes were observed in the mRNA expression of *Il6*, *Il1b*, *Tnfa* (data not shown) in any gut segments, a common signature was found in the duodenum, jejunum, ileum, and distal colon of *db/db* mice as compared to *ob/ob* mice, which were characterized by a significant increase of the expression of *Rorgt2*, a transcription factor that directs the differentiation of inflammatory T helper (Th)17 cells [22]. Moreover, in the duodenum, ileum and distal colon, an up-regulation of *Il17a* (also known as Il17), one of the major pro-inflammatory cytokines secreted by the Th17 cells, was also observed in the *db/db* mice (Figure 1B). Despite an equal body weight and fat mass observed in both mutant mice [19], these results highlight a different inflammatory profile in the small and large intestine of the *ob/ob* and *db/db* mice that can be dissociated from an obese state.

### 3.2. Different eCBome Profiles in the Duodenum of ob/ob and db/db Mice

We have shown that an altered eCBome signaling contributes to the different inflammatory tone in the gut [13]. We also reported that the distinct inflammation in the liver and adipose tissue of *db/db* vs. *ob/ob* mice was accompanied by divergent alterations in eCBome mediator levels as well as eCBome-related receptors and metabolic enzymes [20]. Therefore, we investigated the eCBome tone in the duodenum, jejunum, ileum and distal colon of both mutant mice. We focused here on the duodenum and distal colon, which presented most of the observed modifications in eCBome-related mediators, receptors and anabolic/catabolic enzymes. The results for the jejunum and ileum, which presented only mild changes, are presented as Appendix A, respectively, and are not described below for the purposes of brevity. We will discuss hereafter only the alterations that were significantly different between *ob/ob* and *db/db* mice and that might underlie the observed differences in inflammation-related indicators. In the duodenum (Figure 2A), we did find a statistically significant increase, in *db/db* with respect to *ob/ob* mice, of the duodenal concentration of the 2-monoacylglycerols, particularly 2-palmitoylglycerol (2-PG), 2-linoleoylglycerol (2-LG), 2-arachidonoylglycerol (2-AG), 2-eicosapentaenoylglycerol (2-EPG), 2-docosapentaenoylglycerol (2-DPG) and 2-docosahexaenoylglycerol (2-DHG). The levels of docosapentaenoic acid (DPA), an omega-3 fatty acid which is the precursor of 2-DPG, were also augmented. Conversely, as shown in (Figure 2B), we observed a statistically significant decrease, in *db/db* compared to *ob/ob* mice, of the *N*-acylethanolamines, i.e., *N*-palmitoylethanolamine (PEA), *N*-oleoylethanolamine (OEA), *N*-linoleylethanolamine (LEA) and anandamide (AEA). The levels of an endogenous putative metabolite of AEA, *N*-arachidonoyl glycine (NAGly), were also significantly decreased in the *db/db* group (Figure 2C), whereas the levels of 13-HODE and 13-HODE-G, the oxidized derivatives of linoleic acid (LA) and 2-LG, respectively, were significantly higher in *db/db* compared to *ob/ob* mice (Figure 2C). A significant reduction was also observed for the levels of the arachidonic acid-derived oxylipin, 11-HETE in *db/db* mice as compared to *ob/ob* mice. Despite the fact that the concentration of the omega-3 fatty acids (i.e., the essential alpha linolenic acid, ALA, the eicosapentaenoic acid, EPA, and the docosahexaenoic acid, DHA) did not differ between the *ob/ob* and *db/db* groups (data not shown), the levels of their bioactive derivatives 13(S)-HOTre, 12- and 15-HEPE, and 14 and 17-HDHA, respectively, were significantly higher in *db/db* than *ob/ob* mice (Figure 2C). We also examined the duodenal concentrations of non-eCBome mediators, such as the prostaglandins, and found a significant decrease of PGD_2_ and PGF_2α_ levels in *db/db* with respect to *ob/ob* mice (Figure 2D). We then investigated the modulation of the mRNA expression of genes encoding for eCBome-related receptors or anabolic and catabolic enzymes, as this could explain in part the changes found in the levels of the eCBome mediators. Regarding the receptors (Figure 3A), there was a significant difference only in the expression of the orphan G-protein-coupled receptors *Gpr18*, which was reduced in the duodenum of *db/db* with respect to *ob/ob* mice, in agreement with what was observed with the proposed GPR18 ligand, NAGly. Although not statistically significant (*p* = 0.06), the same trend was observed for *Trpa1*, a proposed target for the eCB, AEA. Differences in gene expression were also observed at the level of eCBome-related metabolic enzymes (Figure 3B). 

Specifically, for the 2-monoacylglycerol metabolic enzymes, the most relevant results were found for the biosynthetic *Plcb1* (*p* = 0.056) and *Pla1a*, and for the catabolic *Abhd16a* (*p* = 0.06) and *Ces1d* whose transcripts were less expressed in *db/db* compared with *ob/ob* mice. The transcript levels of *Dgke*, which is an intracellular lipid kinase that phosphorylates diacylglycerol (DAG) to phosphatidic acid, were also decreased in *db/db* mice. Concerning NAE metabolic enzymes, there were decreases, in *db/db* compared to *ob/ob* mice, of the transcript levels of NAE biosynthetic enzymes *Gde1* and *Napepld*, the latter showing only a trend (*p* = 0.06) toward reduction, which could explain the decreased levels of several NAEs, as described above. However, an increased expression was observed for the anabolic *Pla2g5*. With regard to NAE hydrolyzing enzymes, the expression of *Faah* was reduced in *db/db* compared to *ob/ob* mice. We also observed an increase in the transcript levels of the lipoxygenase *Alox15* and a decrease of *Ptgs2* (*Cox2*) in the *db/db* mice. The reduction of the expression of *Ptgs2*, which is the rate-limiting enzyme primarily responsible for the conversion of cell membrane-derived arachidonic acid into prostaglandins, could explain the decrease of PGF_2α_ and PGD_2_ levels, but also the increase of 2-AG, which is a good substrate for this enzyme.

### 3.3. Different eCBome/Oxylipin Profiles in the Colon of ob/ob and db/db Mice

Concerning the eCBome-related mediators (Figure 4A), the colon only exhibited an increase of the 2-monoacylglycerol, 2-DPG, in *db/db* as compared to *ob/ob* mice. Accordingly, the levels of DPA were also augmented in the *db/db* group. As for the NAEs, only the eCB and AEA, showed a decreasing trend (*p* = 0.09). Modifications were also observed for the omega 3 derivative molecules; particularly 15-HEPE, 17-HDHA, and 13(S)-HOTre, which are the metabolites of EPA, DHA and ALA, respectively, which were increased in the *db/db* compared to the *ob/ob* group, although no changes were observed in the free omega 3 fatty acid levels. Regarding the eCBome receptors, we observed, a significant increase in the gene expression of *Cacna1b* and *Cacna1h*, *Gpr18,* and *Trpv4* in *db/db* mice (Figure 4B). We also analyzed possible differences in the gene expression levels of eCBome-related metabolic enzymes (Figure 4C). Regarding 2-MAG enzymes, we found an increase in the transcript levels of *Mgll*, *Abhd6*, *Agk*, *Ppt1,* and *Abhd16a*, in *db/db* with respect to *ob/ob* mice. Among the metabolic enzymes for NAE biosynthesis and degradation, in the *db/db* group we observed significant differences for the anabolic *Inpp5d*, *Pla2g5*, *Ptpn22*, all of them displaying an increase in the gene expression levels. The concentration of the lipoxygenase *Alox15* was also augmented in the diabetic mice.

### 3.4. Correlations between eCBome/Oxylipin Mediators and Inflammatory Markers in the Intestine

Given that the duodenum and the distal colon are the two gut segments with the most distinctive eCBome profiles when comparing *ob/ob* and *db/db* mice, we investigated correlations between inflammatory markers and bacterial metabolites (i.e., LPS) and eCBome mediator tissue concentrations or metabolic enzyme and receptor mRNA expression levels of the respective two gut segments. An analysis of the Pearson’s rank correlation matrix confirmed the existence of potential links between certain eCBome lipids and genes and either inflammatory markers or bacterial metabolites. In detail, we found that the Gram-negative bacterial component LPS was significantly correlated with the duodenal levels of the PPARα agonist, and hence the anti-lipogenic and anti-inflammatory mediator, 2-PG (Figure 5A) and the colonic mRNA expression levels of *Abhd16a* (Figure 5B). Additionally, in the site with the highest microbial density (i.e., the distal colon), we observed that the two inflammatory markers (i.e., *Rorgt2* and *Il17a*) were positively correlated with 13S-HoTre and the mRNA expression levels of *Trpv4*, *Mgll*, *Abhd16a*, *Inpp5d*, and *Alox 15* (Figure 5B), respectively. While the former gene encodes for a putative polyunsaturated NAE, 2-MAG and oxylipin cation channel receptor (TRPV4), *Mgll* and *Abhd16a* encode for 2-AG inactivating enzymes, *Inpp5d* for a potential NAE biosynthetic enzyme, and *Alox15* for an oxylipin biosynthetic, and polyunsaturated NAE and 2-MAG converting enzyme. Collectively, these observations suggest that eCBome signaling may be involved in modulating, or being modulated by, various inflammatory markers as well as by bacterial metabolites in two different gut segments, whose functions are closely related to intestinal inflammation and permeability.

### 3.5. Correlations between eCBome/Oxylipin Mediators and Gut Microbiota Taxa

Changes in the composition and functionality of the gut microbiota have been associated with impaired intestinal barrier function resulting in bacterial contact with the intestinal epithelium and inflammation, and the translocation of bacterial components [9]. Therefore, we may not rule out that the shift in certain bacterial taxa observed in *ob/ob* and *db/db* mice could partly underlie, or be caused by, alterations in the eCBome signaling, thereby directly and indirectly contributing to the different intestinal inflammatory tone observed between obese and diabetic mice. To this end, we investigated correlations between eCBome and related mediator tissue concentrations or eCBome metabolic enzyme and receptor mRNA expression levels and the absolute abundance of bacterial taxa that were, or tended to be, significantly different between *ob/ob* and *db/db* mice. 

When exploring such correlations using Spearman’s rank correlation matrix, we found that several bacterial taxa were either positively or negatively correlated with molecules involved in eCBome signaling (Figure 6A,B). In particular, *Clostridium*_*sensu*_*stricto*_1 and *Dubosiella* were both negatively correlated with the duodenal concentrations of DPA, 13-HODE, 13-S-HOTre, 15-HEPE, 14HDHA (Figure 6A). Additionally, *Clostridium*_*sensu*_*stricto*_1 was negatively correlated with the mRNA levels of the NAE anabolic enzyme encoding gene, *Pla2g5*, and *Dubosiella* was negatively and positively correlated with 17-HDHA, a potential anti-inflammatory mediator, and the proposed *N*-acyl-glycine receptor, *Gpr18*, respectively (Figure 6A). *Turicibacter* was negatively correlated with 13-HODE, 13-S-HOTre, 15-HEPE, 14-HDHA, and positively correlated with *Grp18*; *Bacteroides* was negatively correlated with *Gpr18* (Figure 6A). The same bacterial taxa, i.e., *Clostridium*_*sensu*_*stricto*_*1*, *Dubosiella* and *Turicibacter*, were negatively correlated with the colonic concentrations of the potential anti-inflammatory mediators 15-HEPE, 17-HDA, 13S-HoTre, and the 2-MAG catabolic enzyme encoding gene, *Abhd16a* (Figure 6B). *Dubosiella* and *Turicibacter* were both negatively correlated with *Alox15*, which encodes for an enzyme involved in both oxylipin biosynthesis and polyunsaturated NAE and 2-MAG oxidation, and *Turicibater* was negatively correlated with the proposed NAE anabolic enzyme encoding gene, *Ptpn22* (Figure 6B). Taken together, these results emphasize that, although anatomically distant from each other, the duodenum and colon share common correlations between their eCBome profile and the fecal microbiota, further underlying how these two systems can be closely interconnected during a state of inflammation occurring during obesity and T2D.

## 4. Discussion

The eCBome modulates several physiological functions, including GI tract function and energy metabolism [23]. It is also involved in the control of inflammatory states, both at the local level and during the low-grade inflammation that is typical of several chronic conditions, such as metabolic disorders, which are partly due to pathological perturbations of the gut microbial ecosystem balance, known as gut dysbiosis. The latter condition has indeed been repeatedly shown to be associated with, among others, chronic GI disorders and metabolic diseases [3,9]. For this reason, we have investigated here, in relation to the presence of inflammatory biomarkers and previously described alterations of the fecal microbiota, changes in eCBome signaling in two genetic models of obesity, the *ob/ob* and *db/db* mice, the latter of which are characterized by several more features of T2D and increased inflammation in the adipose tissue, although exhibiting body weight and fat mass very similar to *ob/ob* mice [19]. The first important novel finding of this study consisted in the observation of a robust and general increase in the expression of two intestinal inflammatory genes in *db/db* mice in both the small and the large intestine, i.e., *Rorgt2*, a transcription factor that directs the differentiation of inflammatory T helper (Th)17 cells [22], and, accordingly, *Il17a* (also known as Il17), one of the major pro-inflammatory cytokines secreted by Th17 cells. This observation confirms a previous finding of ours [19] that the different inflammatory profile of some tissues from the two types of mice is partly independent of their obese state. The intestinal expression of other examined cytokines, including *Il6*, *Il1b* and *Tnfa*, did not differ between the two mice strains, suggesting a selectively impaired Th17 cell balance and a greater inflammatory tone in the intestine of *db/db* mice. Moreover, the increase in the total intestinal length observed in *db/db* mice might reflect impaired intestinal epithelial proliferation, apoptosis, and mucosal hyperplasia, as previously described in *db/db* mice with the same body weight as *ob/ob* mice [24]. 

More relevant to the aims of this study, we observed that the increased inflammatory response in the intestine of *db/db* mice was accompanied by significant alterations of eCBome mediators and receptors, as well as of other oxylipins and eicosanoids, as compared to *ob/ob* mice. Common to all intestinal segments were the increases of 2-DPG and 13-HOTrE, which, given their biosynthetic origin from omega-3 fatty acids, are expected to have anti-inflammatory properties. These changes could represent protective responses to inflammation, but studies on the pharmacological activity of these compounds are needed to substantiate this hypothesis. Indeed, pro-inflammatory prostaglandins were, instead, decreased in both the duodenum and ileum and/or jejunum of *db/db* mice, and so was the potentially pro-inflammatory oxylipin, 11-HETE. On the other hand, the levels of the TRPV1 agonist, 13-HODE-G [25], were elevated, and those of anti-inflammatory prostaglandins were reduced in the ileum and/or jejunum of *db/db* as compared to *ob/ob* mice. These observations, also substantiated by several other, more segment-specific differences in the levels of other eCBome mediators and oxylipins with proposed opposing functions in inflammation, suggest that the observed increased intestinal inflammatory state of *db/db* mice engenders a series of adaptive and maladaptive responses at the level of these lipid signals, whose effects might also depend on the concomitant, and again segment-dependent, changes of their receptors. In support of this hypothesis are two other novel findings of the present study: (1) the levels of two well established anti-inflammatory mediators, PEA [26] and NAGly [27], were significantly reduced in the duodenum (but not in other intestinal segments) of *db/db* compared to *ob/ob* mice, and (2) while no alteration in the expression of the proposed receptors of the former compound (*Ppara*, *Trpv1* and *Gpr55*) was detected, two of the proposed targets of NAGly, the anti-inflammatory *Gpr18* and the pro-inflammatory *Cana1h*, for which this mediator is an agonist and an inhibitor, respectively [27,28], also underwent changes that were different in the duodenum (a decrease and no change, respectively) and the colon (an increase in both cases). *Trpv4*, another emerging pro-inflammatory receptor [29], which has been suggested to be activated by some arachidonic acid derivatives not analyzed here, as well as by AEA [30], was instead increased in both the jejunum and distal colon. Clearly, in view of the several observed differences between *db/db* and *ob/ob* mice in the levels of the lipid mediators investigated here and the expression of their several known receptors, the assessment of the significance of each of these changes in determining the increased intestinal inflammatory state of diabetic mice will require several further investigations.

We also studied the segment-specific mRNA expression of enzymes catalyzing the biosynthesis and degradation of the lipid mediators analyzed here to gain a preliminary understanding as to whether or not putative differences between *db/db* and *ob/ob* mice could explain, at least in part, the different levels of the mediators. In the duodenum, the main enzymes for both the biosynthesis and inactivation of both 2-MAGs and NAEs were less expressed in *db/db* compared with *ob/ob* mice, offering no explanation as to why the levels of the former mediators were mostly increased, whereas those of the NAEs were mostly decreased. This discrepancy was even stronger in the distal colon (and jejunum), where the expression levels of the catabolic enzymes for 2-MAGs, but not those of their anabolic enzymes, were increased, and yet the concentrations of 2-DPG were also counterintuitively increased. The effect of the availability of the ultimate biosynthetic precursors for these fatty acid derivatives may in part have prevailed on the expression of metabolic enzymes, since we found that in all intestinal segments the levels of DPA, but not those of other measured fatty acids, were increased in *db/db* compared with *ob/ob* mice, thus reflecting the increased concentrations of 2-DPG. On the other hand, the relative levels of expression, between *db/db* and *ob/ob* mice, of *Alox15* and *Ptgs2* (respectively increased and decreased in the former genotype) in the duodenum, and of the former enzyme also in the jejunum and distal colon (again increased in the former genotype in both segments), explained the increased and decreased levels of several 15-LO and COX-2 derivatives in intestinal sections of *db/db* mice. The increased levels of some 15-LO derivatives of linoleic acid were also possibly sustained by the observed concomitant increase of this fatty acid precursor, at least in the jejunum and ileum.

Finally, the observed correlations among eCBome and other lipid mediators and the levels of inflammatory markers or the relative abundance of some gut microbiota taxa reinforce the hypothesis that alterations in the intestinal tone of the eCBome and oxylipin signaling are related to higher inflammation and dysbiosis in *db/db* mice. In particular, the pro-inflammatory bacterial component LPS was significantly correlated with the duodenal levels of 2-PG, a PPARα agonist with likely anti-inflammatory action, whereas, in the distal colon, the two inflammation markers found to be up-regulated in *db/db* mice, i.e., *Rorgt2* and *Il17a*, were positively correlated with the potentially anti-inflammatory mediator, 13S-HoTre, and with the mRNA expression levels, respectively, of the pro-inflammatory channel *Trpv4*, and of the inactivating enzymes of mostly anti-inflammatory 2-MAGs, i.e. *Mgll* and *Abhd16a*, as well with *Alox 15,* which produces both pro-and anti-inflammatory mediators. These correlations suggest that alterations in the signaling of eCBome mediators and oxylipins may depict segment-specific adaptive or maladaptive responses to the increased intestinal inflammation observed in *db/db* mice. Likewise, the correlations observed between the absolute abundance of fecal bacterial taxa involved in gut inflammation [31,32,33,34], i.e., *Clostridium*_*sensu*_*stricto*_*1*, *Dubosiella*, *Bacteroides* and *Turicibacter*, with both duodenal and colonic concentrations of several 15-LO derivatives and *Alox15* and *Gpr18* expression levels, reinforced this hypothesis. This is also supported by the fact that the quantity of these bacterial taxa, which was significantly, or tended to be, affected by either the *ob/ob* or the *db/db* genotype or both, was also either positively or negatively correlated with several metabolic and inflammatory markers in other body sites (e.g., liver and adipose tissue) [19], further supporting the concept that certain bacterial taxa may be implicated in the modulation of host metabolic and inflammatory functions.

In summary, we discovered that *db/db* mice display a higher inflammatory state in the intestine when compared to *ob/ob* mice. This difference is associated with profound and potentially adaptive or maladaptive, and partly intestinal segment-specific, alterations in eCBome and oxylipin signaling. These alterations: (1) are only to a minor extent explained by changes in the expression of the corresponding metabolic enzymes, and (2) correlate with both inflammatory markers and fecal microbiota taxa that have been previously found to be altered during conditions of intestinal inflammation in mice, or following the correction thereof. This study opens the way to future investigations on the biological role of several poorly investigated eCBome mediators and oxylipins in the context of obesity and T2D-induced gut dysbiosis and inflammation.

## Figures and Tables

**Figure 1 cells-12-00411-f001:**
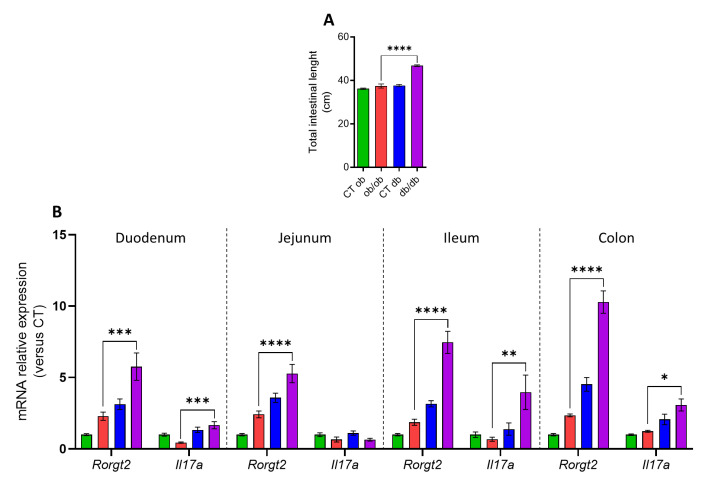
Distinct intestinal length and inflammatory profile in *ob/ob* and *db/db* mice. (**A**) Total intestinal length measured at necropsy (cm) (*n* = 7–8). (**B**) mRNA expression of pro-inflammatory markers measured by RT-qPCR. Data are presented as the mean ± S.E.M of *n* = 7–10. * *p* ≤ 0.05, ** *p* ≤ 0.01, *** *p* ≤ 0.005, **** *p* ≤ 0.001. For mRNA expression, relative units were calculated versus the mean of the CT ob mice values set at 1. Data were analyzed by one-way ANOVA followed by a Tukey’s post hoc test. Abbreviations: see Appendix A.

**Figure 2 cells-12-00411-f002:**
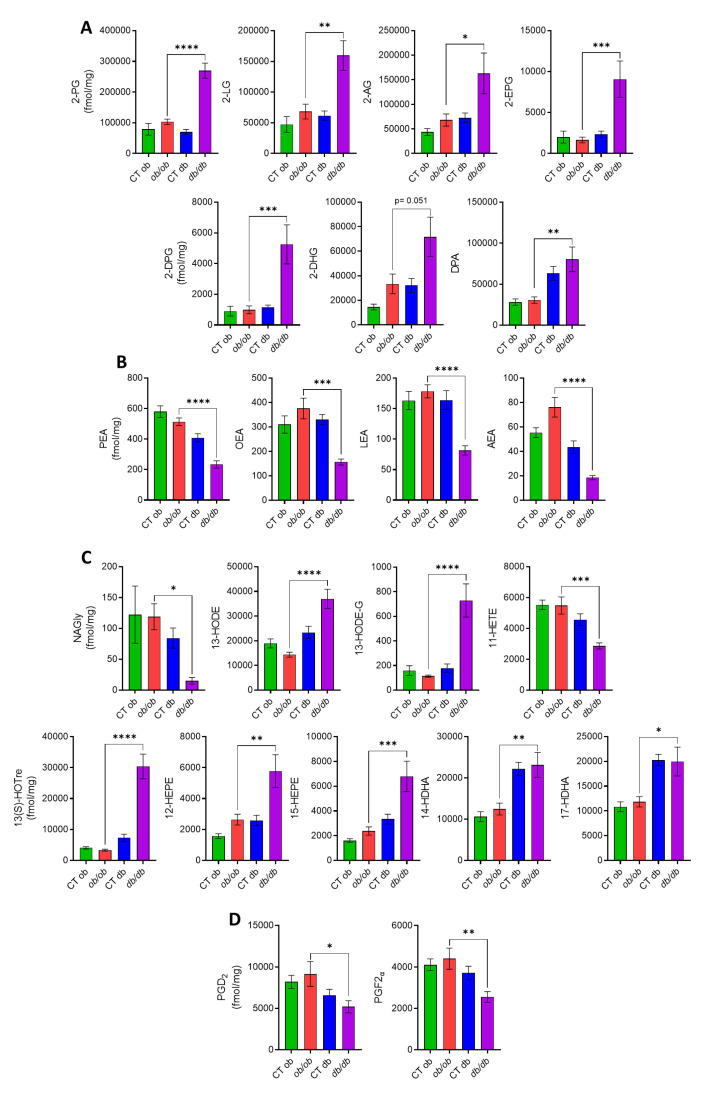
Different eCBome tone in the duodenum of *ob/ob* and *db/db* mice. (**A**–**D**) Concentrations of the eCBome-related mediators in the duodenum (fmol/mg wet tissue weight) measured by HPLC-MS/MS. Data are presented as the mean ± S.E.M of *n* = 7–10. * *p* ≤ 0.05, ** *p* ≤ 0.01, *** *p* ≤ 0.005, **** *p* ≤ 0.001.

**Figure 3 cells-12-00411-f003:**
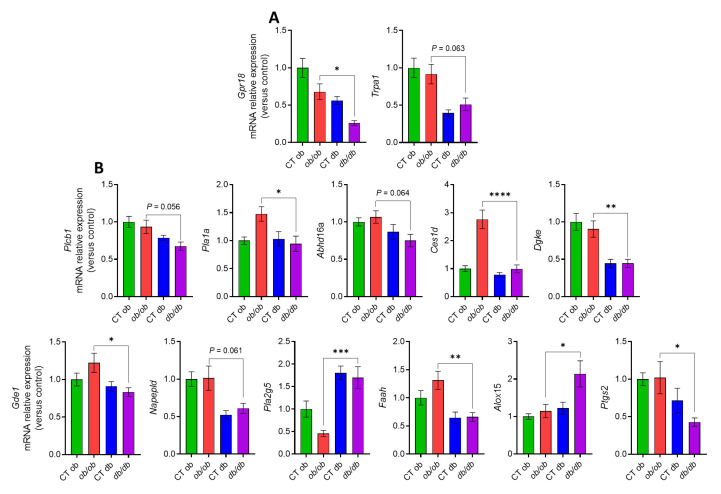
Different mRNA expression, in the duodenum of *ob/ob* and *db/db* mice, of eCBome related receptors (**A**) and metabolic enzymes (**B**) for 2-monoacylglycerols and *N*-acylethanolamines measured by qPCR-based TaqMan Open Array. Data are presented as the mean ± S.E.M of *n* = 7–10. * *p* ≤ 0.05, ** *p* ≤ 0.01, *** *p* ≤ 0.005, **** *p* ≤ 0.001. Relative units were calculated versus the mean of the CT ob mice values set at 1. Data were analyzed by one-way ANOVA followed by Tukey’s post hoc test. Abbreviations: see Appendix A.

**Figure 4 cells-12-00411-f004:**
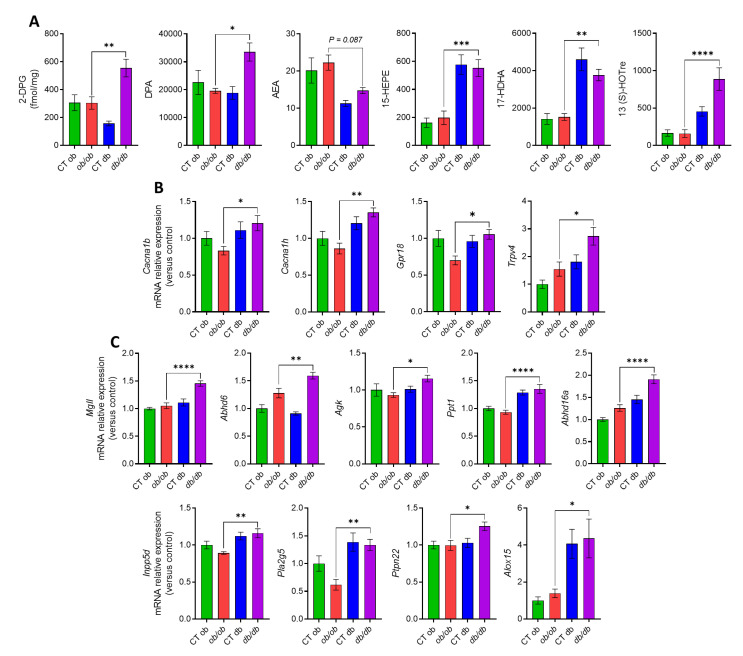
Different eCBome tone in the distal colon of *ob/ob* and *db/db* mice. (**A**) Concentrations of the eCBome-related mediators in the colon (fmol/mg wet tissue weight) measured by HPLC-MS/MS. (**B**) mRNA expression of receptors and (**C**) metabolic enzymes for 2-monoacylglycerols and *N*-acylethanolamines measured by qPCR-based TaqMan Open Array. Data are presented as the mean ± S.E.M of *n* = 7–10. * *p* ≤ 0.05, ** *p* ≤ 0.01, *** *p* ≤ 0.005, **** *p* ≤ 0.001. For mRNA expression, relative units were calculated versus the mean of the CT ob mice values set at 1. Data were analyzed by one-way ANOVA followed by Tukey’s post hoc test. Abbreviations: see Appendix A.

**Figure 5 cells-12-00411-f005:**
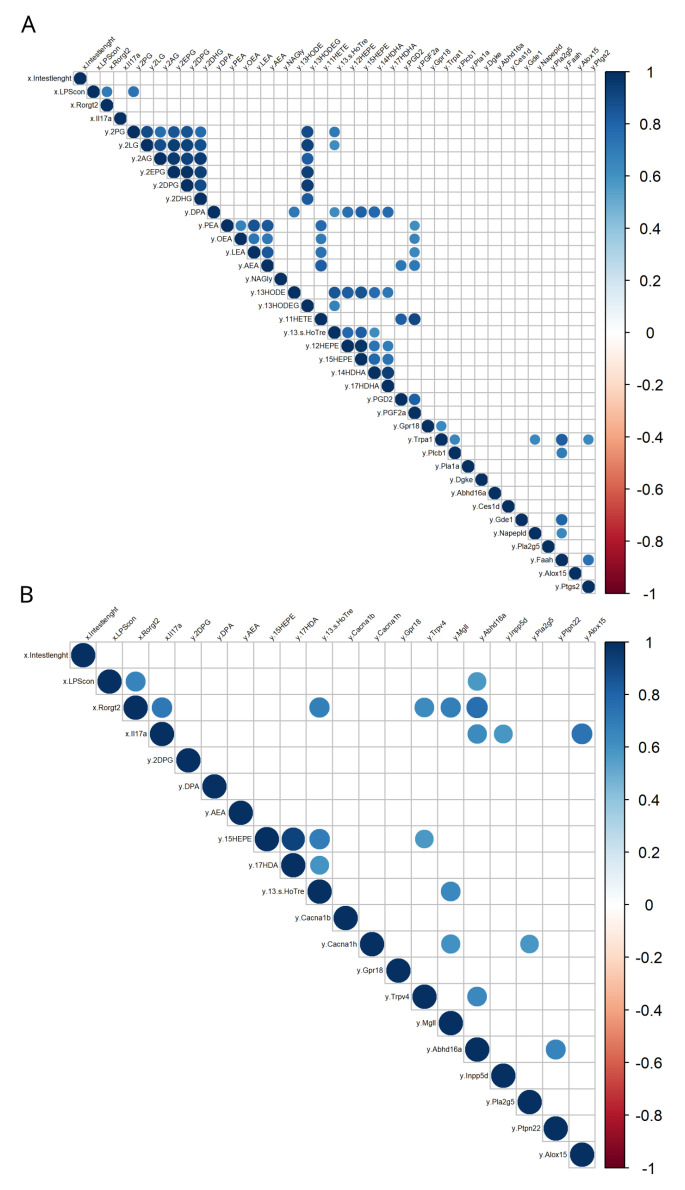
Correlation plot between intestinal length, bacterial components, mRNAs of inflammatory markers, and the eCBome-related mediators measured in two gut segments (i.e., the duodenum and distal colon). (**A**) Correlation matrix showing Pearson correlations with Bonferroni’s adjustment in the duodenum; (**B**) Correlation matrix showing Pearson correlations with Bonferroni’s adjustment in the colon. Positive correlations are displayed in blue and negative correlations in red. Color intensity and size of the circles are proportional to the correlation coefficients. “x” refers to the first data set, the intestinal length, bacterial components, and inflammatory markers, while “y” refers to the second data set, the eCBome-related mediators and mRNAs measured in the two respective gut segments.

**Figure 6 cells-12-00411-f006:**
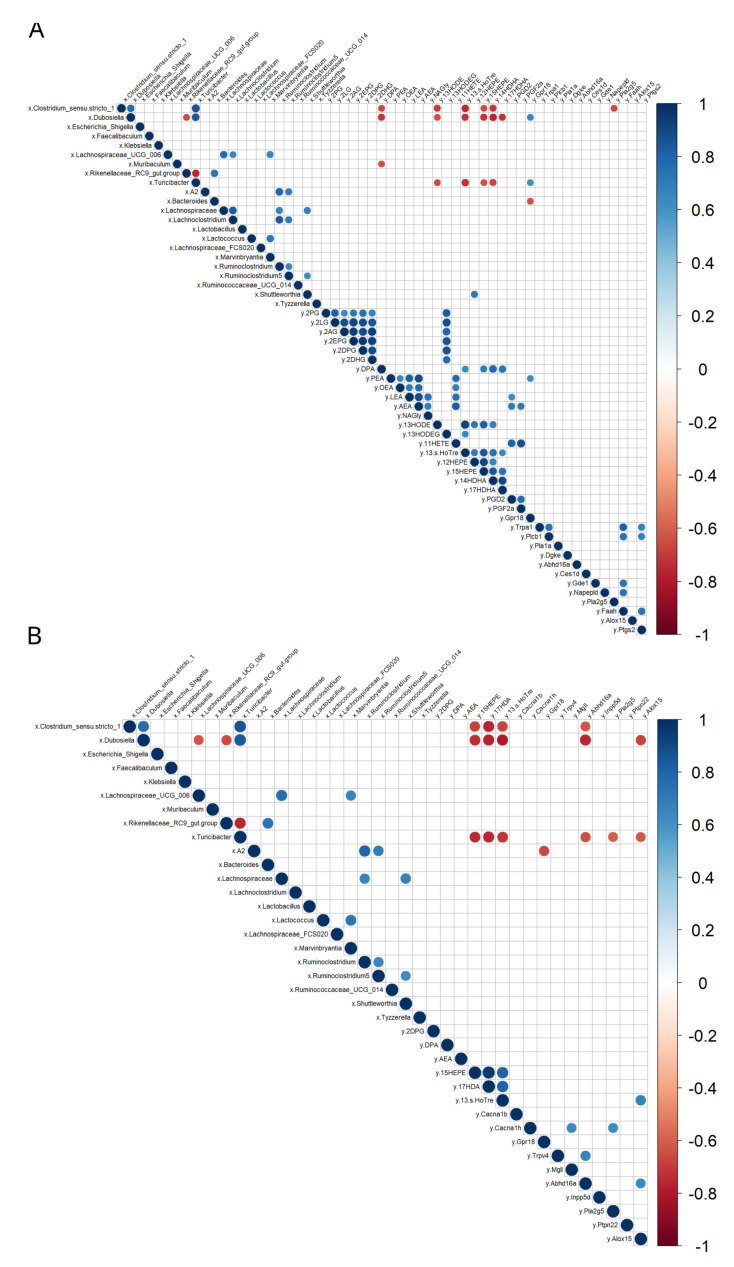
Correlation plot between altered fecal bacterial taxa and eCBome-related mediators measured in two gut segments (i.e., the duodenum and distal colon). (**A**) Correlation matrix showing Spearman correlations with Holm’s adjustment in the duodenum; (**B**) Correlation matrix showing Spearman correlations with Holm’s adjustment in the distal colon. Positive correlations are displayed in blue and negative correlations in red. Color intensity and size of the circles are proportional to the correlation coefficients. “x” refers to the first data set, the fecal bacterial taxa, while “y” refers to the second data set, the eCBome-related mediators and mRNAs measured in the two respective gut segments.

## Data Availability

Data are showed within the manuscript and in the supplemental information files. For the correlation analysis between the eCBome signaling and the gut microbiota, we re-used the microbial data previously published in Suriano et al., [25]. The raw amplicon sequencing data are available in the European Nucleotide Archive (ENA) at EMBL-EBI under accession number PRJEB44809.

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
