# Peer review of "A Lipidomics- and Transcriptomics-Based Analysis of the Intestine of Genetically Obese (ob/ob) and Diabetic (db/db) Mice: Links with Inflammation and Gut Microbiota"

_cells, 2023, doi:10.3390/cells12030411_

Round 1

Reviewer 1 Report

This manuscript by Suriano et al. characterized the profile of endocannabinoidome, eCBome, in ob/ob and db/db mice. They demonstrated that the intestines of db/db mice exhibit more inflammatory phenotypes with a longer intestine. These were associated with the changes in the eCBome-related factors in a segment-dependent manner, which partly explains the metabolic features differentiating ob/ob and db/db. Overall, this is an interesting manuscript. There are just a few issues that need to be clarified, as indicated below.

1.     Please describe the age and sex of the mice. Those factors may affect the metabolic features of the mice and the profiles of their gut microbiota, thus, should be considered.

2.     The title of Results 3.2. (line 226) is the same as the one in Results 3.1.

Reviewer 2 Report

Over all interesting and useful study. My concerns are stated below.

1. Clearer information on sample size is missing here or has not come clearer to me. §135 Lipids were extracted from tissue samples.... but how many?  

2.  In Figure 1A how many samples were there in each CT ob, ob/ob, CT db, db/db? 

3.  Figure 1 A shows, significant total intestinal length difference between ob/ob and db/db has been observed. Now digging into different segments like Duodenum, .., colon as shown in Figure 1B are we observing intestinal length difference?

In my opinion the intestinal length difference effect should be removed before performing following statistical tests for all other omics. 

4. Graphical abstract would be really helpful for readers. 

5. §288 indicates n=7-10. One way ANOVA was applied but please mention which exact statistical test was used and justify?

Reviewer 3 Report

This manuscript is kind of interesting. This study brings lipidomics- and transcriptomics-based analysis of the intestine of genetically obese (ob/ob) and diabetic (db/db) mice and some insight into the biological role of several poorly investigated eCBome mediators and oxylipins in the context of obesity and diabetes-induced gut dysbiosis and inflammation. However, I still have some concerns:

1.      Please take care of your text. The first line of the Abstract showed that Obesity and related metabolic diseases are associated with Low-grade inflammation

2.      In figure 1A, the authors should showed the picture of the intestinal length.

3.      Even this paper has an interesting topic, the lack of normal control mice compare to db/db or ob/ob makes a obvious shortage to analysis and make sure this changes in lipidomics and transcription level.

4.      The author only showed the correlations, is there any type lipids or genes can transform to physiology function?

5.      What is the difference of gut microbiota between db/db and ob/ob, how they contributed the different phenotype?

Round 2

Reviewer 2 Report

Reviewers comment has been addressed by authors as per best of their abilities. 

S.

Reviewer 3 Report

I have no more concerns.